# A High-Gain and High-Efficiency Photovoltaic Grid-Connected Inverter with Magnetic Coupling

**DOI:** 10.3390/mi13101568

**Published:** 2022-09-21

**Authors:** Chien-Hsuan Chang, Chun-An Cheng, Hung-Liang Cheng, En-Chih Chang

**Affiliations:** Department of Electrical Engineering, I-Shou University, Dashu District, Kaohsiung City 84001, Taiwan

**Keywords:** high gain, high efficiency, inverter, magnetic coupling

## Abstract

Conventional photovoltaic (PV) grid-connected systems consist of a boost converter cascaded with an inverter, resulting in poor efficiency due to performing energy processing twice. Many pseudo DC-link inverters with single energy processing have been proposed to improve system efficiency and simplify circuits. However, their output voltage gain is limited by the non-ideal characteristics of the power diode, making them difficult to apply in high-output voltage applications. This paper proposes combining a boost converter with magnetic coupling and a full-bridge unfolding circuit to develop an inverter featuring high voltage-gain and high efficiency. According to the desired instantaneous output voltage, the high-gain boost converter and the full-bridge unfolding circuit are sequentially and respectively controlled by SPWM. A sinusoidal output voltage can be generated by performing energy processing only once, effectively improving the conversion efficiency. Magnetic coupling is adopted to increase the voltage gain of step-up, and the step-down function is realized by the full-bridge unfolding circuit to reduce conduction loss. Finally, a 500 W prototype was fabricated for the proposed high-gain inverter. The experimental results were used to verify the correctness of the theoretical analysis and the feasibility of the circuit structure.

## 1. Introduction

Recently, air pollution has become increasingly serious due to the high consumption of fossil fuels. To reduce carbon dioxide emissions to mitigate global warming and climate change, many researchers are committed to developing renewable energy sources such as photovoltaic (PV), wind, hydro, geothermal and biogas [1,2,3]. Among them, PV energy is attracting increasing attention, and is widely used around the world. There are three types of PV power systems: grid-connected, stand-alone, and hybrid [4,5,6], in which grid-connected systems are the most popular. The PV grid-connected system converts the direct current (DC) of solar energy into alternating current (AC) and feeds it into the grid [7,8].

Due to the low voltage of the PV panels, a low-frequency transformer needs to be added after the inverter in order to be connected with utility, as shown in Figure 1a. However, the low-frequency transformer significantly increases the size and cost of this PV power system. The transformer-less PV grid-connected system is the alternative structure, as shown in Figure 1b. It uses a boost converter to step-up the PV voltage and then converts it to AC power for connection with the utility [9,10]. The transformer-less structure has the advantages of small size and low cost, but its efficiency is reduced because of multiple energy processing stages.

Several single-stage inverters derived from boost or buck converters have been proposed to improve the efficiency [11,12,13], but their application is limited by the need for multiple input sources and the inability to cover a wide range of input voltage variations. Therefore, many pseudo DC-Link inverters have been proposed to overcome these drawbacks [14,15,16,17,18]. Figure 2 shows the block diagram of the pseudo DC-Link inverter, in which a DC/DC converter with both step-up and step-down capabilities is used to generate a rectified sine wave with unipolarity, and an unfolding circuit is cascaded to switch the polarity and to obtain a sinusoidal output voltage. Since the unfolding circuit switches only with the frequency of utility line, a pseudo DC-link inverter that requires only one energy processing can effectively improve conversion efficiency. However, the step-down function is achieved by connecting an additional power switch in series, resulting in higher conduction loss. In addition, the output voltage gain is limited because of the non-ideal characteristics of the power diode, making it unusable for low input voltage or high output voltage applications.

Based on the above considerations, this paper proposes a high-gain and high-efficiency inverter with magnetic coupling, the block diagram of which is shown in Figure 3. The proposed inverter combines a high-gain boost converter with coupling inductor and a full-bridge unfolding circuit. When the instantaneous output voltage is higher than the input voltage, the high-gain boost converter is controlled by sinusoidal pulse-width modulation (SPWM), and the full-bridge unfolding circuit is only used for switching the polarity of the output voltage. When the instantaneous output voltage is lower than the input voltage, the power switch of the high-gain boost converter remains in off state, and the full-bridge unfolding circuit is controlled by SPWM to step down the input voltage to the desired output voltage. This control method is called partial SPWM (P-SPWM), because the high-gain boost converter and the full-bridge unfolding circuit are sequentially and respectively controlled by high-frequency SPWM as the instantaneous output voltage varies. Generally, the following features of the proposed inverter are:Because the high-gain boost converter and the full-bridge unfolding circuit perform high-frequency switching at different times, only one energy processing stage is required to generate the sinusoidal output voltage, which can effectively improve the conversion efficiency.Magnetic coupling is adopted to increase the voltage gain of step-up so that the proposed inverter can be operated with utility of high voltage [19,20].The full-bridge unfolding circuit is used to realize the step-down function so that additional series power switch is not required, which can reduce conduction losses.The proposed inverter has both step-up and step-down capabilities, making it suitable for applications with a wide range of input voltage variations.

## 2. Circuit Configuration

Figure 4 shows the circuit configuration of the proposed high-gain inverter with magnetic coupling, in which *V_DC_* is the input voltage, and *v_o_*(*t*) is the output voltage. In this circuit, the first stage is a high-gain DC-DC boost converter that is controlled by P-SPWM to generate a rectified sine wave with unipolarity. It is formed mainly by the power switch *S_Bo_*, the diode *D_Bo_*, and the coupled inductor. The second stage is a full-bridge unfolding circuit that is controlled by P-SPWM to accomplish step-down function and switches the polarity of the rectified sine wave during step-up mode. The unfolding circuit is mainly composed of the switches *S_Bu_*_1_, *S_Bu_*_2_, *S_Bu_*_3_, and *S_Bu_*_4_, and a low-pass filter formed by the inductor *L*_f_ and the capacitor *C_f_*.

The proposed inverter can be operated in either step-up or step-down mode, depending on the levels of the input and output voltages. In step-down mode, the switch *S_Bo_* remains in off state, and the diode *D_Bo_* remains in the on state. The switches *S_Bu_*_1_ and *S_Bu_*_2_ are controlled by high-frequency SPWM and switch complementarily. The switches *S**_Bu_*_3_ and *S**_Bu_*_4_ switch complementarily with line frequency. Additionally, the low-pass filter is used to filter out the high-frequency components of the output voltage.

When the desired output voltage is higher than the input voltage, the proposed inverter enters step-up mode. In this mode, the switch *S_Bo_* switches with the high-frequency SPWM control. The switches *S_Bu_*_1_, *S_Bu_*_4_ maintain conductance to transfer energy to the output side during the positive half-cycle, and the switches *S_Bu_*_2_, *S_Bu_*_3_ maintain cut-off. In the negative half-cycle, the roles of the switches *S_Bu_*_1_, *S_Bu_*_4_ and the switches *S_Bu_*_2_, *S_Bu_*_3_ are swapped so that the output voltage polarity can be switched.

The high-gain DC-DC boost converter and the full-bridge unfolding circuit do not perform high-frequency switching at the same time, which means that only one energy processing stage is required to convert a low DC voltage into the required AC voltage. Therefore, the conversion efficiency of the proposed inverter can be effectively improved.

## 3. Operation Principles

In this section, the detailed operating principles of the proposed inverter are addressed; the following assumptions are made to simplify the circuit analysis:All circuit elements are ideal.The circuit operates in steady state.Inductor currents are continuous.The dead time of power switches is extremely short and can be ignored.

Figure 5 presents the timing diagram of the proposed high-gain inverter within one cycle of output voltage *v_o_*(*t*), where *V_M_* is the amplitude of *v_o_*(*t*), *T_o_* is the period of *v_o_*(*t*), and *d_Bo_*(*t*), *d_Bu_*_1_(*t*), and *d_Bu_*_3_(*t*) are the duty ratios of switches *S_Bo_*, *S_Bu_*_1_, and *S_Bu_*_3_. Additionally, *V_GS,Bo_*(*t*), *V_GS,Bu_*_1_(*t*), *V_GS,Bu_*_2_(*t*), *V_GS,Bu_*_3_(*t*), and *V_GS,Bu_*_4_(*t*) are the conceptual gate-driving signals of all power switches.

The output voltage *v_o_*(*t*) is sinusoidal and can be divided into positive and negative half-cycles. When the input voltage *V_DC_* is higher than the absolute value of the instantaneous output voltage *v_o_*(*t*), the inverter operates in step-down mode; otherwise, the inverter operates in step-up mode. The operation principles of the negative half-cycle are the same as those of the positive half-cycle, except that the driving signals of switches *S_Bu_*_1_ and *S_Bu_*_2_ are swapped and the driving signals of switches *S_Bu_*_3_ and *S_Bu_*_4_ are swapped. In the following, the operation principles of the proposed inverter are illustrated with respect to the positive half-cycle.

### 3.1. Step-Down Mode

During the positive half-cycle (0 < *t* < *T_o_/*2), the switch *S_Bu_*_4_ keeps turning on, and the switch *S_Bu_*_3_ keeps turning off. When the input voltage *V_DC_* is higher than the output voltage *v_o_*(*t*), the inverter operates in step-down mode. In this mode, the switch *S_Bo_* keeps turning off, and its duty ratio *d_Bo_*(*t*) is zero. The coupled inductors *L_P_* and *L_S_* are connected in series as an input filter. The full-bridge unfolding circuit is controlled by unipolar SPWM. The gate driving signals of the switches *S_Bu_*_1_, *S_Bu_*_2_ are complementary, and their duty ratios can be expressed as follows:(1)$$d_{Bu1}(t)=\frac{V_{M}\cdot \sin\omega t}{V_{DC}},$$
(2)$$d_{Bu2}(t)=1-\frac{V_{M}\cdot \sin\omega t}{V_{DC}},$$
within one switching period, when *S_bu_*_1_ is turned on, and *S_Bu_*_2_ is turned off, the input voltage *V_DC_* simultaneously charges the inductor *L_f_* and provides the energy required for the output load through the diode *D_Bo_*. The equivalent circuit is shown in Figure 6a. When *S_Bu_*_1_ is turned off, and *S_Bu_*_2_ is turned on, the inductor *L_f_* releases energy to the output load. The equivalent circuit is shown in Figure 6b.

Moreover, during the negative half-cycle (*T_o_/*2 < *t* < *T_o_*), the switches *S_Bu_*_3_ and *S_Bu_*_4_ exchange their operation states, and the switch *S_Bo_* keeps turning off. The input voltage *V_DC_* charges the inductor *L_f_* and provides energy to the output load when the switch *S_Bu_*_2_ turns on, and the equivalent circuit is shown in Figure 7a. When the switch *S_Bu_*_2_ is turned off and the switch *S_Bu_*_1_ is turned on, the inductor *L_f_* releases energy to the output load. Figure 7b shows the equivalent circuit.

### 3.2. Step-Up Mode

When the instantaneous output voltage *v_o_*(*t*) is higher than the input voltage *V_DC_*, the proposed inverter enters step-up mode. During the positive half-cycle (0 < *t* < *T_o_/*2), the switches *S_Bu_*_1_ and *S_Bu_*_4_ keep turning on, and the switches *S_Bu_*_2_ and *S_Bu_*_3_ keep turning off. The switch *S_Bo_* operates with high-frequency SPWM control. Figure 8 shows the current waveforms of the coupled inductor operating in continuous current mode (CCM), in which *T* is the switching period. At the initial time *t* = 0, the switch *S_Bo_* turns on to force the diode *D_Bo_* to be off, and the capacitor *C_o_* provides energy for the output load. Figure 9a shows the equivalent circuit of the switch *S_Bo_* turning on.

The voltage across the primary winding of the coupled inductor, *v_LP_*, is equal to the input voltage *V_DC_*, and can be expressed as follows:(3)$$v_{LP}{=V}_{DC}{=L}_{P}\frac{{di}_{LP}}{dt}$$

According to Equation (3), the amount of current change in the primary inductor *L_P_* during the on state of the switch *S_Bo_* can be expressed as
(4)$$\Delta i_{LP}{}_{\left(close\right)}=\frac{V_{DC}\cdot d_{Bo}T}{L_{P}},$$
where *d_Bo_* is the duty ratio of *S_Bo_*. In Figure 8, *i_LP_*(0^+^) is the initial value of the primary inductor current, so the end value *i_LP_*(*d_Bo_T*^−^) of the on state of the switch *S_Bo_* can be expressed as
(5)$$i_{LP}(d_{Bo}T^{-})=i_{LP}(0^{+})+\frac{V_{DC}\cdot d_{Bo}T}{L_{P}},$$

Assuming that the coupled inductor is an ideal element (coupling coefficient *k* = 1), the mutual inductance *M* can be expressed as follows:(6)$$M=\sqrt{L_{P}L_{S}},$$

In addition, if the turn ratio between primary and secondary windings is defined as 1/*N*, the relationship between primary inductor *L_P_* and secondary inductor *L_S_* can be expressed as follows:(7)$$L_{S}=N^{2}L_{P},$$

Substituting Equation (7) into Equation (6) yields the following equation:(8)$$M=NL_{P},$$

At the time t = *d_Bo_T*^−^, the switch *S_Bo_* turns off, and the diode *D_Bo_* is forced into the on state. The primary inductor *L_P_* is connected in series with the secondary inductor *L_S_* to release energy. The coupled inductor and the input voltage *V_DC_* simultaneously transfer energy to the output and charge the capacitor *C_o_*, as shown in Figure 9b. Since the primary and the secondary inductor currents are equal, the primary inductor current *i_LP_*(*d_Bo_T^+^*) after switch *S_Bo_* turns off can be determined by the law of energy conservation as follows:(9)$$i_{LP}(d_{Bo}T^{+})=i_{LS}(d_{Bo}T^{+})=\frac{1}{(1+N)}\times i_{LP}(d_{Bo}T^{-}),$$

When the switch *S_Bo_* is in the off state, the voltage across the coupled inductor can be expressed as
(10)$$v_{LP}+v_{LS}=(V_{DC}-v_{o}(t))=(L_{P}+L_{S}+2M)\times \frac{di_{LP}}{dt},$$

Substituting Equations (7) and (8) into Equation (10), it can be simplified as follows:(11)$$(V_{DC}-v_{o}(t))={(1+N)}^{2}L_{P}\times \frac{di_{LP}}{dt},$$

During the time between *d_Bo_T* and *T*, the inductor current decreases linearly due to the negative voltage across the coupled inductor in Equation (11). The amount of inductor current change during this time interval can be expressed as follows:(12)$$\Delta i_{LP}{}_{\left(open\right)}=\Delta i_{LS}{}_{\left(open\right)}=\frac{(V_{DC}-v_{o}(t))\cdot (1-d_{Bo})\times T}{{\left(1+N\right)}^{2}\times L_{P}},$$

From Equations (9) and (12), the minimum current on the primary winding of the coupled inductor *i_LP_*(*T*^−^) can be expressed as follows:(13)$$i_{LP}(T^{-})=i_{LP}(d_{Bo}T^{+})+\Delta i_{LP}{}_{\left(open\right)}=\frac{i_{LP}(d_{Bo}T^{-})}{(1+N)}+\frac{(V_{DC}-v_{o}(t))\cdot (1-d_{Bo})\times T}{{\left(1+N\right)}^{2}L_{P}},$$

Substituting Equation (5) into Equation (13), the following equation is obtained:(14)$$i_{LP}(T^{-})=\frac{1}{(1+N)}\left[i_{LP}(0^{+})+\frac{V_{DC}\cdot d_{Bo}T}{L_{P}}\right]+\frac{(V_{DC}-v_{o}(t))(1-d_{Bo})T}{{\left(1+N\right)}^{2}L_{P}},$$

At the time *t* = *T_S_*, the switch *S_Bo_*_1_ turns on, and the primary inductor current can be obtained by the law of energy conservation as follows:(15)$$i_{LP}(T^{+})=(1+N)\times i_{LP}(T^{-}),$$

In steady-state operation, since the current *i_LP_*(*T*^+^) is equal to *i_LP_*(0^+^), the output voltage gain can be obtained by substituting Equation (15) into Equation (14) as follows:(16)$$\frac{v_{o}\left(t\right)}{V_{DC}}=\frac{1+Nd_{Bo}\left(t\right)}{1-d_{Bo}\left(t\right)},$$

By expressing the output voltage *v_o_*(*t*) as *V_M_*·sin*ωt*, the duty ratio of the switch *S_Bo_* in step-up mode can be obtained as follows:(17)$$d_{Bo}\left(t\right)=\frac{V_{M}\cdot \sin\omega t-V_{DC}}{V_{M}\cdot \sin\omega t+NV_{DC}},$$

Moreover, when the proposed inverter operates in step-up mode during the negative half-cycle (*T_o_/*2 < *t* < *T_o_*), the switches *S**_Bu_*_1_, *S**_Bu_*_4_ are in the off state, and the switches *S_Bu_*_2_, *S_u_*_3_ are in the on state. The switch *S_Bo_* keeps switching with high frequency, and the operation principles are the same as those for the positive half-cycle. The equivalent circuits for when switch *S_Bo_* is turning on and off are shown in Figure 10a,b, respectively.

According to the analysis above, the status of all switching elements is listed in Table 1. It can be clearly understood that the proposed inverter has only two power elements switching with high frequency in both step-down and step-up modes; therefore, switching losses can be reduced to improve the efficiency.

## 4. Design Considerations

To explain how to determine the component parameters, the design considerations of the coupled inductor, the semiconductor components, and the output filter are addressed in this section.

### 4.1. Boundary Condition of the Coupled Inductor

If the coupled inductor operates in boundary conduction mode (BCM), the minimum current on the primary side *i_LP_*(*T*^−^), as shown in Figure 8, will reach zero just before the switch *S_Bo_* enters the next switching cycle. In BCM, the average inductor current *I_LB_* within one high-frequency cycle can be calculated as follows:(18)$$I_{LB}=\frac{1}{2}{\Delta i}_{LP(\mathrm{close})}{\times d}_{Bo}+\frac{1}{2}{\Delta i}_{LP\left(\mathrm{open}\right)}\times (1-d_{Bo})$$

The amount of current increase and decrease on the primary winding in BCM have the following relationship:(19)$${\Delta i}_{LP(\mathrm{open})}=\frac{{\Delta i}_{LP(\mathrm{close})}}{\left(1+N\right)}$$

Substituting Equation (19) into Equation (18) gives
(20)$$I_{LB}=\frac{∆i_{LP(\mathrm{close})}{\;\times \;(1+Nd}_{Bo})}{2(1+N)}$$

Substituting Equation (4) into Equation (20) yields
(21)$$I_{LB}=\frac{V_{DC}{\;\times \;d}_{Bo}{T\;\times \;(1+Nd}_{Bo})}{{2L}_{P}\left(1+N\right)}$$

Since the average current *I_LB_* is equal to the instantaneous input current, the boundary condition *I_o_*_(*B*)_ of the instantaneous output current can be obtained from Equations (16) and (21), and the relationship of power balance as follows:(22)$$I_{o(B)}=\frac{V_{DC}{\;\times \;d}_{Bo}T\;\times \;(1-d_{Bo})}{{2L}_{P}\left(1+N\right)}$$

When the instantaneous output current *i_o_*(*t*) is equal to *I_o_*_(*B*)_, the proposed inverter operates in BCM, and the boundary inductance of the primary winding *L_P_*_(*B*)_ can be obtained as follows:(23)$$L_{P(B)}=\frac{V_{DC}{\;\times \;d}_{Bo}T\;\times \;(1-d_{Bo})}{{2\times i}_{o}\left(t)\times (1+N\right)}$$

### 4.2. Voltage Stresses of the Power Components

When the switch *S_Bo_* turns off in step-up mode, as shown in Figure 9b, the maximum voltage stress on *S_Bo_* occurs at the peak output voltage, and can be expressed as follows:(24)$$V_{ds,\;Bo(\mathrm{Max})}{=V}_{DC}+\frac{V_{M}-V_{DC}}{1+N}$$

When the switch *S_Bo_* turns on, the diode *D_Bo_* is forced to turn off, as shown in Figure 9a. The maximum voltage stress on *D_Bo_* also occurs at the peak output voltage, and can be expressed as follows:(25)$$V_{D-Bo(\mathrm{Max})}{=NV}_{DC}{+V}_{M},$$

The maximum voltage stress on both the capacitor *C_o_* and the capacitor *C_f_* is the peak output voltage *V_M_*, and can be expressed as follows:(26)$$V_{Co(\mathrm{Max})}{=V}_{Cf(\mathrm{Max})}{=V}_{M},$$

According to Figure 9 and Figure 10, the voltage stresses on the switches *S_Bu_*_1_, *S_Bu_*_2_, *S_Bu_*_3_ and *S_Bu_*_4_ are equal to the voltage across the capacitor *C_o_*. Therefore, the maximum voltage stresses on these four switches of the unfolding circuit can expressed as follows:(27)$$V_{ds,\;Bu1(\mathrm{Max})}{=V}_{ds,\;Bu2(\mathrm{Max})}{=V}_{ds,\;Bu3(\mathrm{Max})}{=V}_{ds,\;Bu4(\mathrm{Max})}{=V}_{M}.$$

### 4.3. Current Stresses of the Power Components

When the inverter operates in step-up mode and the switch *S_Bo_* turns on, the current in the primary winding of the coupled inductor rises. When the output voltage *v_o_* reaches the peak *V_M_*, the current stress on the primary winding reaches its maximum, as indicated below:(28)$$I_{LP(\mathrm{Max})}=\frac{V_{M\;}{(1+Nd}_{Bo})}{R(1-d_{Bo})}+\frac{V_{DC}d_{Bo}T}{L_{P}}$$

When the switch *S_Bo_* turns off, the diode *D_Bo_* is forward biased. The primary and secondary windings are connected in series to the discharge, so the maximum current stress of the secondary winding can be known from Equations (9) and (28), as follows:(29)$$I_{LS(\mathrm{Max})}{=I}_{LP(\mathrm{Max})}\times \;\frac{1}{\left(1+N\right)}=\left[\frac{V_{M\;}{(1+Nd}_{Bo})}{R(1-d_{Bo})}+\frac{V_{DC}d_{Bo}T}{L_{P}}\right]\;\times \;\frac{1}{\left(1+N\right)}$$

In step-up mode, the maximum current stress on the switch *S_Bo_*_1_ is the same as that of the primary winding, expressed as follows:(30)$$I_{ds,\;Bo(\mathrm{Max})}{=I}_{LP(\mathrm{Max})}=\frac{V_{M\;}{(1+Nd}_{Bo})}{R(1-d_{Bo})}+\frac{V_{DC}d_{Bo}T}{L_{P}}$$

Additionally, when the diode *D_Bo_* is conducting, its maximum current stress is the same as that of the secondary winding, expressed as follows:(31)$$I_{DBo(\mathrm{Max})}{=I}_{LS(\mathrm{Max})}=\left[\frac{V_{M\;}{(1+Nd}_{Bo})}{R(1-d_{Bo})}+\frac{V_{DC}d_{Bo}T}{L_{P}}\right]\;\times \;\frac{1}{\left(1+N\right)}$$

Assuming that the output high-frequency current ripple can be completely filtered out and ignored, the current stresses of the switches *S_Bu_*_1_, *S_Bu_*_2_, *S_Bu_*_3_, *S_Bu_*_4_, and the inductor *L_S_* are the same as the output current, and their maximum values can be expressed as follows:(32)$$I_{ds,\;Bu1(\mathrm{Max})}{=I}_{ds,\;Bu2(\mathrm{Max})}{=I}_{ds,\;Bu3(\mathrm{Max})}{=I}_{ds,\;Bu4(\mathrm{Max})}{=I}_{Lf(\mathrm{Max})}=\frac{V_{M\;}}{R}$$

### 4.4. Selection of the Output Filter

When the converter operates in step-down mode, the full-bridge unfolding circuit is controlled by SPWM, and the inductor *L_f_* is used for energy storage. The boundary inductance of *L_f_* can be expressed as follows:(33)$$L_{f(B)}=\frac{R\;\times \;(1-d_{Bu1})T}{2}$$

The inverter can operate in the CCM of step-down mode by selecting the inductance of *L_f_* to be greater than the boundary inductance *L_f_*_(*B*)_.

In addition, the inductor *L_f_* and the capacitor *C_f_* are used as a low-pass filter in step-up mode, and its cut-off frequency *f_c_* can be expressed as:(34)$$f_{c}=\frac{1}{2\pi \sqrt{L_{f}C_{f}}}$$

Once the inductance of *L_f_* is determined, the capacitance of *C_f_* can be designed on the basis of Equation (34) according to the desired cut-off frequency.

## 5. Experimental Results

To verify the feasibility of the proposed high-gain inverter, an experimental prototype was built according to the electrical specifications listed in Table 2. The prototype was tested using input voltages between 100 V and 200 V, in order to verify that the proposed inverter is suitable for PV panels with wide-range voltage variations. Additionally, the output voltage was selected as 220 V_rms_ in order to prove the high boosting capacity of the proposed inverter.

On the basis of the output voltage of 220 V_rms_ and the output power of 500 W, the equivalent load resistance (*R*) can be calculated as being 96.8 Ω. By selecting the BCM of step-down mode at an instantaneous output current *i_o_*(*t*) of 0.6 A and an input voltage *V_DC_* of 100 V, the boundary inductance of *L_f_* can be calculated as being about 1 mH using Equation (33). By selecting a cut-off frequency *f_c_* of 5 kHz, the capacitance of *C_f_* can be obtained using Equation (34) as 1 μF.

To ensure that the maximum duty ratio is below 0.5, the turn ratio *N* is chosen as 1.5. At the input voltage of 100 V and the peak output voltage of 312 V, the maximum duty ratio can be calculated using Equation (17) as being around 0.46. Additionally, the condition of peak output current and 40% load is selected to operate in BCM, thus avoiding excessive values of primary inductance and the saturation of the magnetic core. From Equation (23), the boundary inductance of the primary winding *L_P_*_(*B*)_ can be obtained as being about 193 μH. In the actual design, 200 μH is used as the primary inductance, and the secondary inductance of 450 μH can be obtained from Equation (7). Based on the previous design and calculations, the selected component parameters of the experimental prototype are summarized in Table 3.

Figure 11 shows the measured waveforms of the output voltage *v_o_*(*t*) and the output current *i_o_*(*t*) under the condition of 100 V input voltage and 500 W output power. It can be seen that the output voltage and current are both near-ideal sinusoidal waves with low distortion, verifying that the proposed circuit is indeed capable of converting DC input to AC output.

Figure 12 shows the measured waveforms of the gate-driving signals of the switches *S_Bo_*, *S_Bu_*_1_ and *S_Bu_*_2_ under the condition of 100 V input voltage and 500 W output power. When the absolute output voltage |*v_o_*(*t*)| is lower than 100 V, the proposed inverter operates in step-down mode. The switches *S_Bu_*_1_ and *S_Bu_*_2_ switch with high frequency, and the switch *S_Bo_* remains in the off state. Conversely, the proposed inverter operates in step-up mode. The switch *S_Bo_* is switching with high frequency, and the switches *S_Bu_*_1_ and *S_Bu_*_2_ perform low-frequency switching only to switch the polarity of output voltage.

Figure 13 shows the measured waveforms of the currents and the voltages of the coupled inductor. From Figure 13a, the coupled inductor functions as a filter in step-down mode, so that the voltages *v_LP_* and *v_LS_* are almost zero. When the inverter operates in step-up mode, the voltages *v_LP_* and *v_LS_* vary with high-frequency switching of the switch *S_Bo_*. Figure 13b shows the zoomed-in waveforms at the peak of the output voltage *v_o_*(*t*). The inductor current *i_LP_* is operated in CCM, verifying previous theoretical calculations and parametric design. Additionally, the inductor currents *i_LP_* and *i_LS_* are equal during the switch *S_Bo_* turning off, because the primary and secondary inductors discharge in series.

To verify that the proposed inverter is suitable for a wide range of input voltages, the input voltage is increased to 200 V for testing. Figure 14 shows he measured waveforms of the output voltage *v_o_*(*t*) and the output current *i_o_*(*t*) under the condition of 200 V input voltage and 500 W output power. Both the output voltage and current can be maintained in low-distortion sine waves, which proves that the proposed inverter is suitable for a wide range of input voltages. The gate-driving signals at 200 V input voltage are shown in Figure 15. The control strategy is similar to that at 100 V input. Due to the higher input voltage, the time interval becomes longer for the step-down mode and shorter for the step-up mode.

Figure 16 presents the measured waveforms of the currents and the voltages of the coupled inductor. As shown in Figure 16a, the coupled inductor is still used as a filter in step-down mode, and the inductor voltages are almost zero. The waveforms are zoomed-in at the peak of the output voltage *v_o_*(*t*) and shown in Figure 13b. Due to the higher input voltage, the duty ratio of the switch *S_Bo_* is reduced, and the charging time of the primary inductor is shorter. During the switch *S_Bo_* turning off, the inductor currents *i_LP_* and *i_LS_* are still the same, proving that the primary and secondary inductors are discharging in series to increase voltage gain.

The total harmonic distortion (T.H.D.) and odd-order harmonics of the output voltage at full load are measured and listed in Table 4. All measured harmonics comply with the standard of EN6100-3-2 Class C. The efficiency curves of the proposed inverter are illustrated in Figure 17. As can be seen, the conversion efficiencies reach up to 96.1% at 200V input voltage and up to 94.2% at 100V input voltage, verifying that the proposed inverter can indeed achieve high efficiency. Additionally, Figure 18 shows a photograph of the prototype hardware used for the experimental measurements, in which the dsPIC33FJ16GS504 development board is used to generate the P-SPWM driving signals, and a wire-wound resistor is used as a testing load.

To further verify that the proposed inverter has the ability to adjust with utility line voltage fluctuations, Figure 19 shows the experimental waveforms at 230 V_rms_ output and 100 V input. It can be seen that the output voltage is still a near-ideal sinusoidal wave with low distortions. Its measured T.H.D. is 1.75%, which is slightly higher than that of 220 V_rms_ output.

## 6. Conclusions

A high-gain and high-efficiency inverter with magnetic coupling was successfully developed and implemented. The digital signal processor dsPIC33FJ16GS504 was used to generate the gate-driving signals of the proposed inverter, which can simplify the complexity of the control circuit and improve the reliability. As the instantaneous output voltage changes, the proposed circuit sequentially operates in step-down and step-up modes. In each operation mode, only one energy processing is required to obtain the desired output voltage. In addition, to significantly reduce switching losses, conversion efficiency can be effectively improved because part of the energy is delivered directly to the output load. Additionally, by adding a coupled inductor to the boost circuit, the voltage gain of the proposed inverter can be increased, so that it is suitable for applications with low input voltage.

## Figures and Tables

**Figure 1 micromachines-13-01568-f001:**
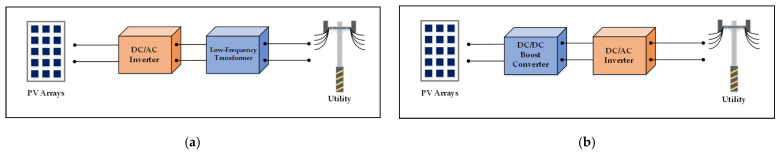
The structures of PV grid-connected power systems with (**a**) a low-frequency transformer; (**b**) a boost converter.

**Figure 2 micromachines-13-01568-f002:**
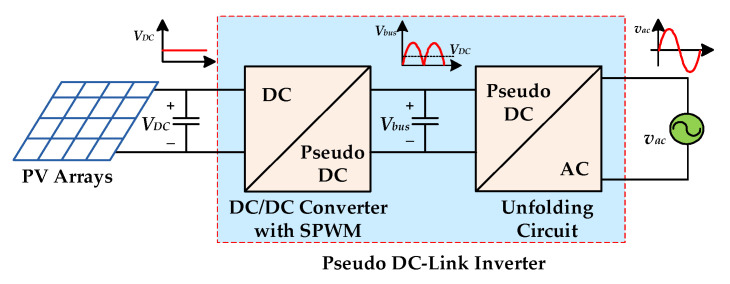
The grid-connected PV power system with a pseudo DC-link inverter.

**Figure 3 micromachines-13-01568-f003:**
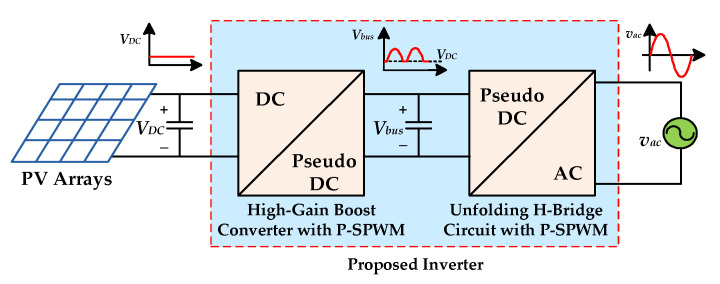
The grid-connected PV power system with the proposed high-efficiency inverter.

**Figure 4 micromachines-13-01568-f004:**
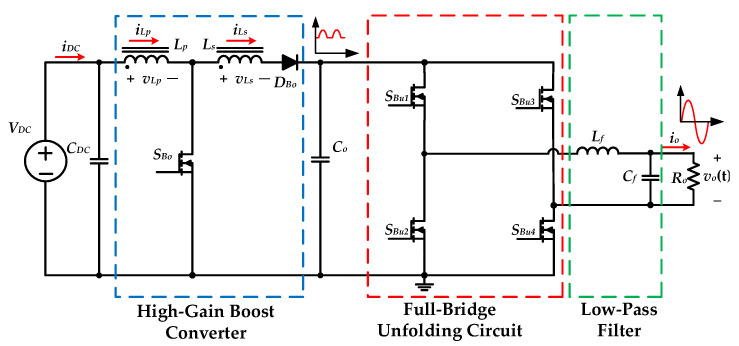
The circuit configuration of the proposed high-gain inverter with magnetic coupling.

**Figure 5 micromachines-13-01568-f005:**
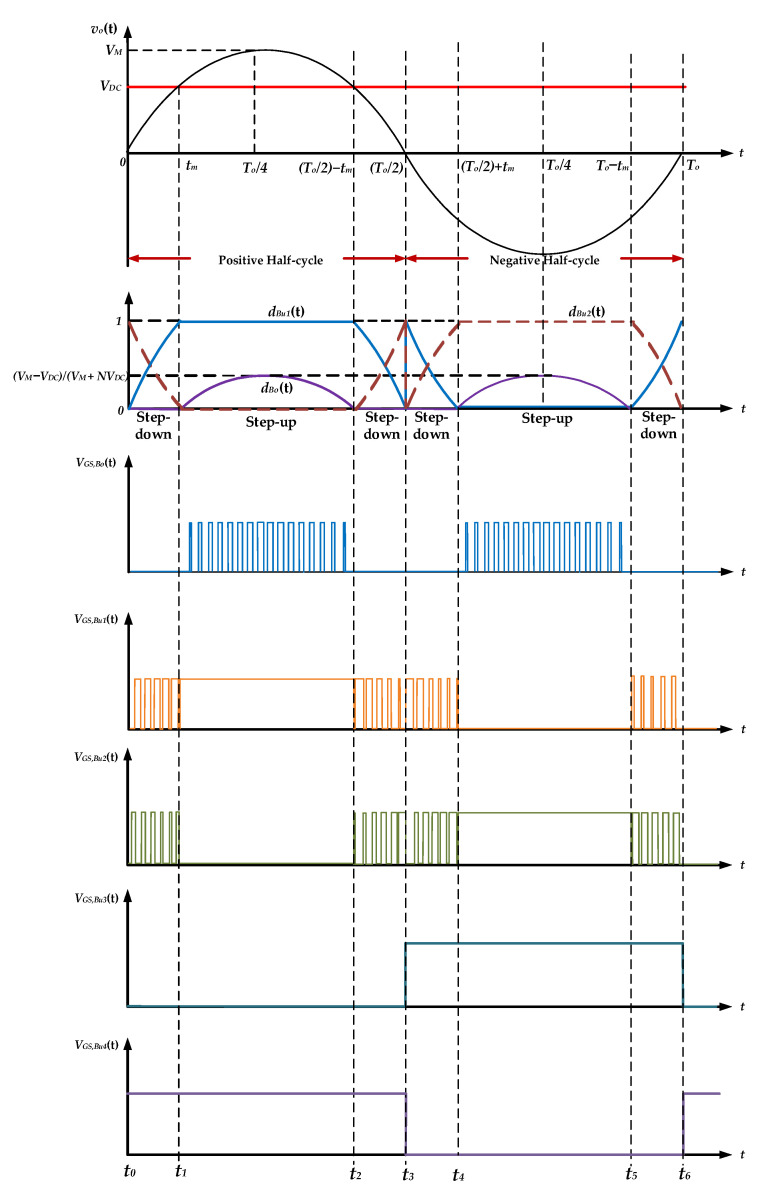
The timing diagram of the proposed high-gain inverter.

**Figure 6 micromachines-13-01568-f006:**
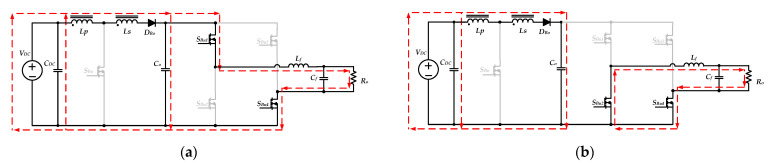
The equivalent circuits of the inverter operating in step-down mode during the positive half-cycle: (**a**) *S_Bu_*_1_ on, *and S_Bu_*_2_ off; (**b**) *S_Bu_*_1_ off, *and S_Bu_*_2_ on.

**Figure 7 micromachines-13-01568-f007:**
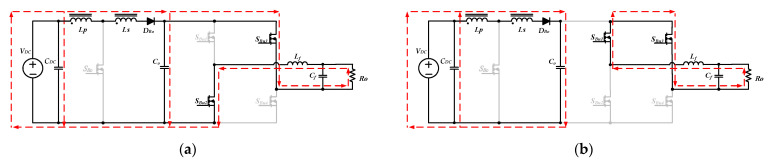
The equivalent circuits of the inverter operating in step-down mode during the negative half-cycle: (**a**) *S_Bu_*_1_ off and *S_Bu_*_2_ on; (**b**) *S_Bu_*_1_ on and *S_Bu_*_2_ off.

**Figure 8 micromachines-13-01568-f008:**
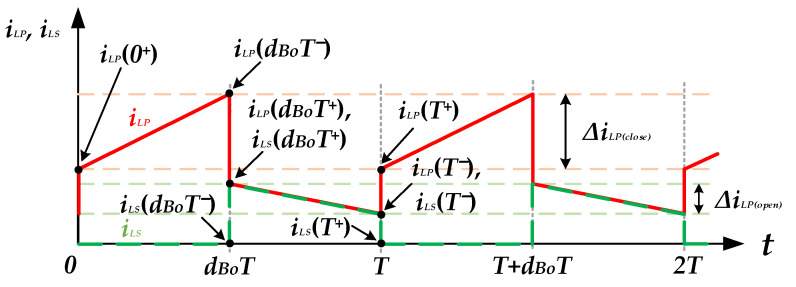
The current waveforms of the coupled inductor operating in CCM.

**Figure 9 micromachines-13-01568-f009:**
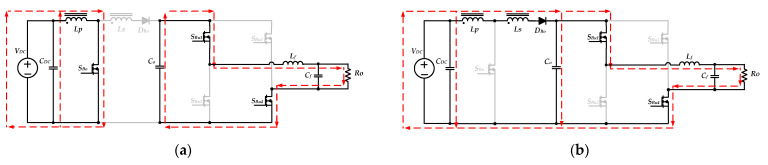
The equivalent circuits of the inverter operating in step-up mode during the positive half-cycle: (**a**) *S_Bo_* on; (**b**) *S_Bo_* off.

**Figure 10 micromachines-13-01568-f010:**
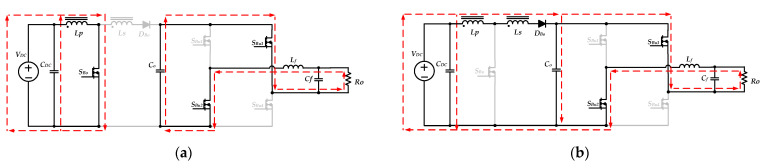
The equivalent circuits of the inverter operating in step-up mode during the negative half cycle: (**a**) *S_Bo_* on; (**b**) *S_Bo_* off.

**Figure 11 micromachines-13-01568-f011:**
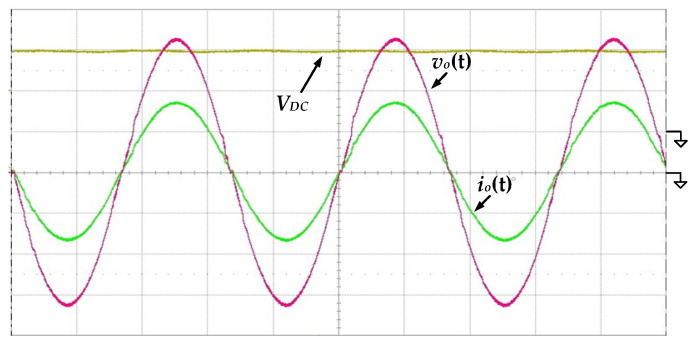
The measured waveforms of the output voltage *v_o_*(*t*) and output current *i_o_*(*t*) at 100 V input voltage and 500 W output load (*v_o_*(*t*): 100 V/div; *i_o_*(*t*): 2 A/div; *V_DC_*: 100 V/div; time: 5 ms/div).

**Figure 12 micromachines-13-01568-f012:**
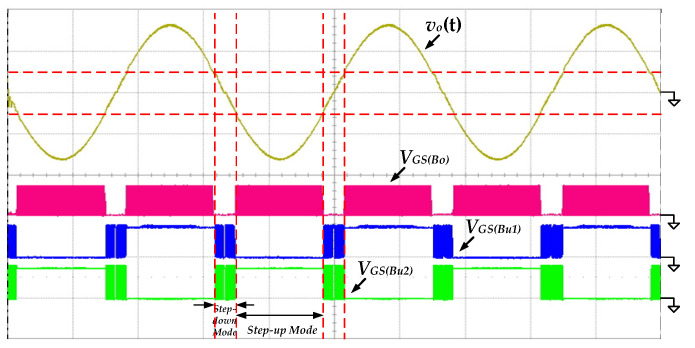
The measured waveforms of the gate-driving signals at 100 V input voltage and 500 W output load (*v_o_*(*t*): 200 V/div; *V_GS_*_(*Bo*)_*, V_GS_*_(*Bu*1)_*, V_GS_*_(*Bu*2)_: 20 V/div; time: 5 ms/div).

**Figure 13 micromachines-13-01568-f013:**
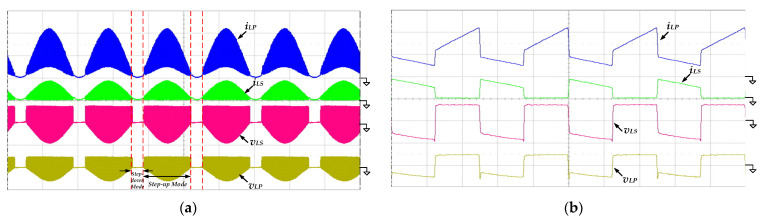
The measured waveforms of the inductor currents *i_LP_*, *i_LS_* and the inductor voltages *v_LP_* and *v_LS_* at 100 V input voltage, 500 W output load and (**a**) low-frequency line cycle (*v_L_**_P_*, *v_L_**_S_*: 200 V/div; *i_LP_*, *i_LS_*: 10 A/div; time: 5 ms/div); (**b**) high-frequency switching cycle (*v_L_**_P_*, *v_L_**_S_*: 200 V/div; *i_LP_*, *i_LS_*: 10 A/div; time: 20 μs/div).

**Figure 14 micromachines-13-01568-f014:**
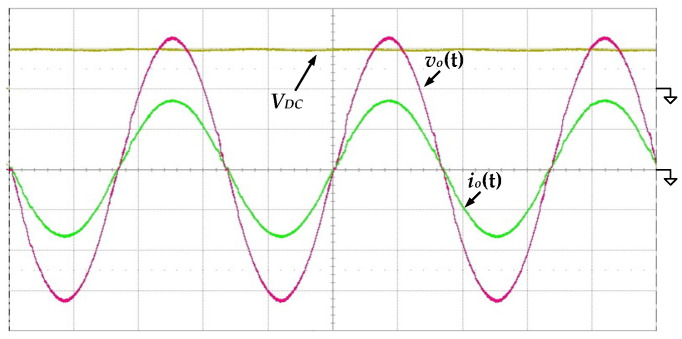
The measured waveforms of the output voltage *v_o_*(*t*) and output current *i_o_*(*t*) at 200 V input voltage and 500 W output load (*v_o_*(*t*): 100 V/div; *i_o_*(*t*): 2 A/div; *V_DC_*: 200 V/div; time: 5 ms/div).

**Figure 15 micromachines-13-01568-f015:**
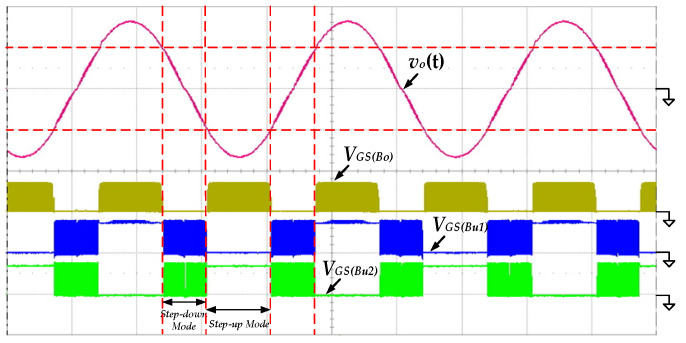
The measured waveforms of the gate-driving signals at 200 V input voltage and 500 W output load (*v_o_*(*t*): 200 V/div; *V_GS_*_(*Bo*)_*, V_GS_*_(*Bu*1)_*, V_GS_*_(*Bu*2)_: 20 V/div; time: 5 ms/div).

**Figure 16 micromachines-13-01568-f016:**
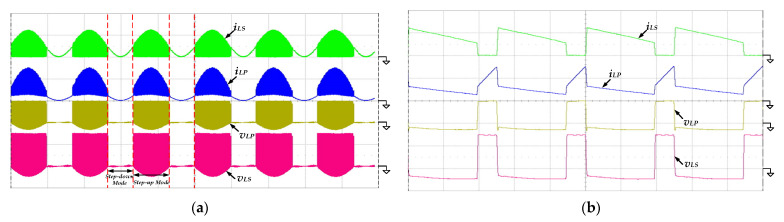
The measured waveforms of the inductor currents *i_LP_*, *i_LS_* and the inductor voltages *v_LP_* and *v_LS_* at 200 V input voltage, 500 W output load and (**a**) low-frequency line cycle (*v_L_**_P_*, *v_L_**_S_*: 200 V/div; *i_LP_*: 10 A/div; *i_LS_*: 5 A/div; time: 5 ms/div); (**b**) high-frequency switching cycle (*v_L_**_P_*, *v_L_**_S_*: 200 V/div; *i_LP_*: 10 A/div; *i_LS_*: 5 A/div; time: 20 μs/div).

**Figure 17 micromachines-13-01568-f017:**
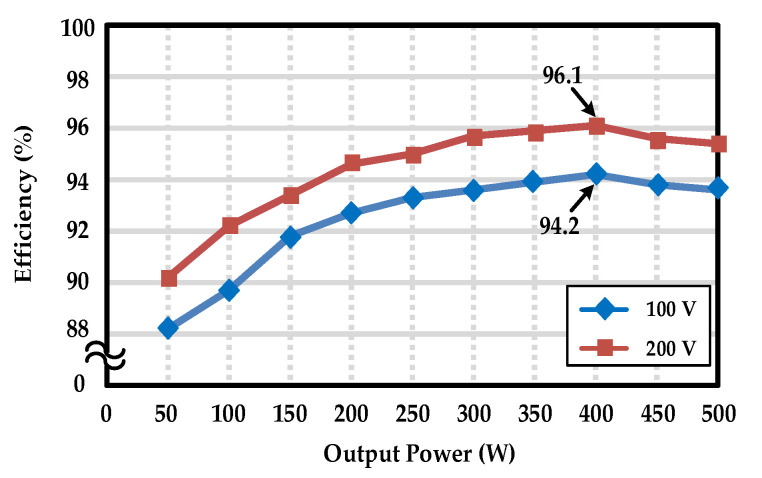
Measured efficiency curves of the proposed inverter.

**Figure 18 micromachines-13-01568-f018:**
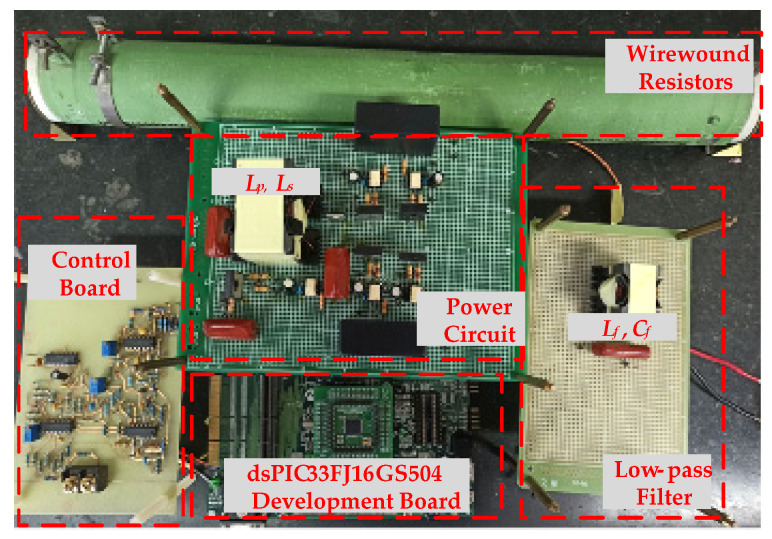
Photograph of the experimental prototype.

**Figure 19 micromachines-13-01568-f019:**
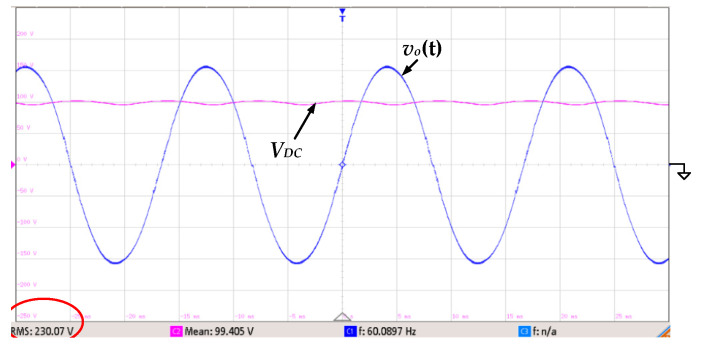
The experimental waveforms of the output voltage *v_o_*(*t*) and input voltage *V_DC_* at 230 V_rms_ output and 100 V input (*v_o_*(*t*): 100 V/div; *V_DC_*: 50 V/div; time: 5 ms/div).

**Table 1 micromachines-13-01568-t001:** Status of switching elements of the proposed high-gain inverter.

Element	Positive Half-Cycle (0 < *t* < *T_o_/*2)		Negative Half-Cycle (*T_o_/*2 < *t* < *T_o_*)	
	Step-Down Mode	Step-Up Mode	Step-Down Mode	Step-Up Mode
*S* * _Bu_ * _1_	Switching with *d_Bu_*_1_(*t*)	Always on	Switching with (1 − *d_Bu_*_1_(*t*))	Always off
*S_Bu_* _2_	Switching with (1 − *d_Bu_*_1_(*t*))	Always off	Switching with *d_Bu_*_1_(*t*)	Always on
*S_Bu_* _3_	Always off	Always off	Always on	Always on
*S_Bu_* _4_	Always on	Always on	Always off	Always off
*S_Bo_*	Always off	Switching with *d_Bo_*(*t*)	Always off	Switching with *d_Bo_*(*t*)
*D* * _Bo_ *	Always on	Switching with (1 − *d_Bo_*(*t*))	Always on	Switching with (1 − *d_Bo_*(*t*))

**Table 2 micromachines-13-01568-t002:** Electrical specifications of the proposed high-gain inverter.

Electrical Specifications	
Input voltage, *V_DC_*	100–200 (V)
Output voltage, *v_o_*	220 (V_rms_)
Line frequency, *f_o_*	60 (Hz)
Output power, *P_o_*	500 (W)
Switching frequency, *f*	20 (kHz)
Switching period, *T*	50 (μs)

**Table 3 micromachines-13-01568-t003:** Component parameters of the proposed high-gain inverter.

Component Parameters	
MOSEET, *S**_Bo_*	SPW47N60C3 (650 V/47 A)
MOSEETs, *S**_Bu_*_1_, *S**_Bu_*_2_, *S**_Bu_*_3_ and *S**_Bu_*_4_	IRFP460 (500 V/20 A)
Diode, *D**_Bo_*	C4D10120A (1200 V/14 A)
Turn Ratio, *N*	1.5
Primary Inductance, *L_P_*	200 μH
Secondary Inductance, *L_S_*	450 μH
Capacitor, *C_o_*	1 μF
Inductor, *L_f_*	1 mH
Capacitor, *C_f_*	1 μF

**Table 4 micromachines-13-01568-t004:** The total harmonic distortion and odd-order harmonics of the output voltages.

Harmonic	100 V	200 V
T.H.D.	1.73%	1.13%
3rd Harmonic	1.57%	0.87%
5th Harmonic	0.36%	0.29%
7th Harmonic	0.24%	0.33%
9th Harmonic	0.11%	0.17%
11th Harmonic	0.07%	0.12%

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
