# Peer review of "A High-Gain and High-Efficiency Photovoltaic Grid-Connected Inverter with Magnetic Coupling"

_micromachines, 2022, doi:10.3390/mi13101568_

Round 1
Reviewer 1 Report
26 to developing renewable energy 26 sources such as photovoltaic (PV), wind, hydro, geothermal and biogas [1-3]. Among 27 them, PV energy is the most mature and is widely used around the world.
hydro has been in use much longer than PV. Hydro has been in use more than 100 years and it is more mature than PV technology.
Tests are provided for output voltage 220V rms. How about THD if you chose 230 V rms as an output voltage. Is it small difference or higher. I am asking because your converter is close to saturation. Could it operate at 230V rms output? Please comment this.
Author Response
We wish to thank the reviewer for your time and valuable comments on our paper. We have made necessary changes requested by the reviewers. As a result, the revised manuscript will be much more valuable. In the revised manuscript, the revised parts according to your comments are highlighted in yellow. We would like further explain our responses as follows:
Comments to the Author
Point 1: 26 to developing renewable energy 26 sources such as photovoltaic (PV), wind, hydro, geothermal and biogas [1-3]. Among 27 them, PV energy is the most mature and is widely used around the world.
hydro has been in use much longer than PV. Hydro has been in use more than 100 years and it is more mature than PV technology.
Response 1: Thank you for your helpful comments. We revised "the most mature" to "growing attention".
Please refer to page 1, line 27 in the revised manuscript.
Point 2: Tests are provided for output voltage 220V rms. How about THD if you chose 230 Vrms as an output voltage. Is it small difference or higher. I am asking because your converter is close to saturation. Could it operate at 230V rms output? Please comment this.
Response 2: Thank you for your helpful comments. The experimental result of 230 Vrms output has been included in the revised manuscript. There is no saturation issue. THD is 1.75% and is small higher than that of 220 Vrms output.
Please refer to page 15, line 367 and Figure 19 in the revised manuscript.

Reviewer 2 Report
Dear authors
I appreciate your support by selecting the Micromachines Journal for possible publication of your research work.
The paper develops a new inverter with high efficiency based on a combination between a boost converter with magnetic coupling and a full-bridge unfolding circuit .
The paper is well structured, combining a literature research, mathematical models and experimental rests, all these leading to valuable results, so I consider that the manuscript can be accepted for publication.
Best regards.
Author Response
We wish to thank the reviewer for your time and valuable comments on our paper. We have made necessary changes requested by the reviewers. As a result, the revised manuscript will be much more valuable. In the revised manuscript, the revised parts according to your comments are highlighted in yellow. We would like further explain our responses as follows:
Comments to the Author
Point 1: I appreciate your support by selecting the Micromachines Journal for possible publication of your research work.
The paper develops a new inverter with high efficiency based on a combination between a boost converter with magnetic coupling and a full-bridge unfolding circuit.
The paper is well structured, combining a literature research, mathematical models and experimental rests, all these leading to valuable results, so I consider that the manuscript can be accepted for publication.
Response 1: Thank you for your kind comments.
